# A Rapid-Patterning 3D Vessel-on-Chip for Imaging and Quantitatively Analyzing Cell–Cell Junction Phenotypes

**DOI:** 10.3390/bioengineering10091080

**Published:** 2023-09-13

**Authors:** Li Yan, Cole W. Dwiggins, Udit Gupta, Kimberly M. Stroka

**Affiliations:** 1Fischell Department of Bioengineering, University of Maryland, College Park, MD 20742, USA; cdwiggin@umd.edu (C.W.D.); ugupta13@terpmail.umd.edu (U.G.); 2Biophysics Program, University of Maryland, College Park, MD 20742, USA; 3Center for Stem Cell Biology and Regenerative Medicine, University of Maryland, Baltimore, MD 21201, USA; 4Marlene and Stewart Greenebaum Comprehensive Cancer Center, University of Maryland, Baltimore, MD 21201, USA

**Keywords:** blood-brain barrier, 3D vessel-on-chip, tight junctions, cell morphology

## Abstract

The blood-brain barrier (BBB) is a dynamic interface that regulates the molecular exchanges between the brain and peripheral blood. The permeability of the BBB is primarily regulated by the junction proteins on the brain endothelial cells. In vitro BBB models have shown great potential for the investigation of the mechanisms of physiological function, pathologies, and drug delivery in the brain. However, few studies have demonstrated the ability to monitor and evaluate the barrier integrity by quantitatively analyzing the junction presentation in 3D microvessels. This study aimed to fabricate a simple vessel-on-chip, which allows for a rigorous quantitative investigation of junction presentation in 3D microvessels. To this end, we developed a rapid protocol that creates 3D microvessels with polydimethylsiloxane and microneedles. We established a simple vessel-on-chip model lined with human iPSC-derived brain microvascular endothelial-like cells (iBMEC-like cells). The 3D image of the vessel structure can then be “unwrapped” and converted to 2D images for quantitative analysis of cell–cell junction phenotypes. Our findings revealed that 3D cylindrical structures altered the phenotype of tight junction proteins, along with the morphology of cells. Additionally, the cell–cell junction integrity in our 3D models was disrupted by the tumor necrosis factor α. This work presents a “quick and easy” 3D vessel-on-chip model and analysis pipeline, together allowing for the capability of screening and evaluating the cell–cell junction integrity of endothelial cells under various microenvironment conditions and treatments.

## 1. Introduction

The blood-brain barrier (BBB) is key to central nervous system health. The BBB is composed of brain microvascular endothelial cells (BMECs) lining the cerebral vessels, along with a host of supporting neural cells, such as pericytes, astrocytes, and glial cells. The BBB structure is responsible for separating the brain tissue from the contents of the brain’s blood supply [1]. The key molecular structures responsible for the BBB’s high selectivity are the tight junctions and adherens junctions in the brain’s endothelial cells [2]. The degeneration of these tight junctions puts the brain at risk. For example, in Alzheimer’s Disease and other neurodegenerative disorders, BBB breakdown is linked to the onset and progression of the disease [3]. As such, understanding what factors contribute to the loss of tight junction integrity in the BBB is key to understanding how neurodegenerative diseases arise, and what risk factors contribute to disease [4].

Developing in vitro models of the BBB is imperative for discovering the mechanisms underlying BBB breakdown [4,5]. A number of approaches have already been developed to model the BBB and study transport mechanisms across BMECs, both in vivo and in vitro [6]. While in vivo models are accurate at mimicking the BBB environment, they involve heavy use of animals and often lack relevance to humans [7]. For example, in in vivo BBB modeling studies involving intravenous injection monitoring, each data point collected requires the use of one animal, and there are inherently high degrees of variability present between animals [8]. On the other hand, in vitro techniques avoid the use of animals and lot-to-lot variability, but creating an accurate representation of the complex BBB environment is challenging. Simple models of the BBB have been developed by using cell culture systems such as the side-by-side diffusion chambers and Transwell culture wells, either of which can promote the coculture of BMECs and other BBB adaptors cells (e.g., astrocytes and/or pericytes) [9,10,11,12]. These methods are particularly effective at assessing the mechanisms and kinetics of the transendothelial transport of various molecules of interest. Furthermore, the protein expression of requisite BMEC markers, such as efflux transporters, has been very well characterized in multiple BMEC culture models [13,14].

More recently, various approaches have been developed to analyze BBB barrier function [5,6]. Engineered microvessel platforms have integrated BMECs inside a microfluidic channel coated with extracellular matrix proteins (ECM), such as collagen. These engineered vessels demonstrate permeability changes in response to circulating permeation factors, and these changes are often similar to responses seen in the clinic [15]. The advantage of using microfluidic approaches is that microenvironmental cues, such as matrix stiffness, composition, shear stress, geometry, and the presence of biochemical factors, can all be studied independently and with precise control, as opposed to an in vivo model where the systematic and controlled investigations of these microenvironmental cues are not possible [16,17,18,19,20,21].

Transport processes, both transcellular (across/through cells) and paracellular (between cells) have been well-studied using in vitro BBB models [22]. However, much less is known about how microenvironment changes impact BMEC cell–cell junction integrity, which is the major contributing factor to BBB paracellular permeability [23,24]. One reason for this lack of knowledge is that there are not many techniques that can quantitatively and rigorously assess how cell–cell junction phenotypes change in various microenvironmental conditions. To address this need, we recently developed a novel Python-based Junction Analyzer Program (JAnaP) that provides a comprehensive, quantitative analysis of the presentation of cell–cell junctions in 2D fluorescence images of cell monolayers immunostained for cell–cell junction proteins. We have used the JAnaP to detect changes in cell–cell junction presentation in response to various microenvironment conditions, including matrix stiffness, cell culture conditions, tumor cell-secreted factors, rhinovirus C infections, and novel light-based drug delivery mechanisms [25,26,27,28,29,30].

Three-dimensional microvessel models are becoming more common and are demonstrating high potential as models of the BBB [6]; hence, there is a growing need for the ability to quantitatively assess cell–cell junctions not only in 2D monolayers but also in 3D structures. For example, 3D microvessel models of human brain microvascular endothelial cells (HBMECs) and human umbilical vein endothelial cells (HUVECs) display morphological differences when cultured on cylindrical glass rods varying in diameter and when exposed to varying degrees of shear stress [19]. In that study, images of the 3D microvessels were obtained using confocal microscopy, and a custom MATLAB-based “UNWRAP” program was used to convert the 3D confocal image stacks into 2D surfaces to analyze cell morphology [19].

Our study aimed to establish a proof-of-concept 3D microvessel model and cell–cell junction analysis pipeline that could be used for future applications related to the BBB or microvessel models of other vascular beds. We developed a fast (i.e., requiring 3 h of labor) and simple (i.e., able to be completed even without microfabrication facilities) protocol that creates 3D microvessels with polydimethylsiloxane and stainless-steel acupuncture needles. The 3D microvessels were seeded with human-induced pluripotent stem cell-derived brain microvascular endothelial cells, immunostained for cell–cell junction proteins, and imaged in 3D via confocal microscopes. The 3D image stacks were converted to 2D surfaces using the aforementioned UNWRAP program and subsequently analyzed using our JAnaP. Using this method, we were able to quantify the cell–cell junction phenotypes, cell morphology, and cell size of the iBMEC-like cells in the 3D microvessel devices and demonstrate that these parameters are sensitive to treatment with barrier-reducing conditions, such as TNF-α treatment. These results establish the feasibility of this method for future in vitro studies of the BBB and have the potential to impact the biomedical engineering industry by allowing engineers to streamline drug development testing or develop methods for earlier BBB disease diagnosis.

## 2. Materials and Methods

### 2.1. Cell Culture

Human induced pluripotent stem cells (iPSCs) (DF19-9-11 T.H; WiCell, Madison, WI, USA) were cultured on Matrigel (Corning, Oneonta, NY, USA) in E8 medium (Thermo Fisher Scientific, Waltham, MA, USA) and subsequently differentiated into induced brain microvascular endothelial (iBMEC)-like cells as previously described [10]. Initially, the iPSCs were treated with Accutase (Thermo Fisher Scientific) and plated onto Matrigel-coated 6-well plates at a density of 1.5 × 10^5^ cells/cm^2^ in E8 medium supplemented with 10 μM Y27632 (R&D Systems). The day after seeding, the E6 medium (Thermo Fisher Scientific) was introduced to initiate differentiation and changed daily thereafter. Day 0 indicates the time of initiating the differentiation in E6 medium. On day 4, the EC culture medium was applied, which includes human endothelial serum-free medium (Thermo Fisher Scientific), 1% platelet-poor plasma-derived serum (Thermo Fisher Scientific), 20 ng/mL basic fibroblast growth factor (bFGF; Peprotech, Cranbury, NJ, USA), and 10 mM retinoic acid (RA; Sigma, St. Louis, MO, USA). On day 6, the cells were dissociated with Accutase and subcultured into the ECM-coated microvessel devices prepared as described above. On day 7, the EC medium was replaced with EC medium without RA and bFGF for maintenance purposes.

### 2.2. Vessel-on-Chip Fabrication

Microfluidic chips were made using polydimethylsiloxane (PDMS) (Dow Silicones Corporation, Midland, MI, USA), within which a ø200 µm microneedle (Seirin Corporation, Shizuoka, Japan) was embedded. PDMS was made first by using a 10:1 ratio by mass of Sylgard 184 Silicone Elastomer Base to 184 Silicone Elastomer Base Curing Agent. The components were mixed and poured onto a silicon wafer within a 150 mm Petri dish (VWR, Radnor, PA, USA). One day before making the PDMS mix, the silicon wafers were silanized using tridecafluoro-1,1,2,2, tetrahydrooctyl-1-trichlorosilane (OTS, 97%) (Sigma) overnight in a vacuum desiccator. The PDMS was placed in a vacuum desiccator for 20 min to remove air bubbles. Once degassed, the PDMS was placed in an 80 °C oven for approximately 1 h. The cured PDMS was then diced into 20 mm × 25 mm rectangular chips with an X-ACTO knife and peeled from the silicon wafer. A 5 mm × 15 mm rectangular section was then cut in the center of the chip and two slits were made at the top of the chip along the bisection line. The microneedle was then inserted into the slits, and the chip was flipped over and placed back onto the wafer. Uncured PDMS was then carefully poured into the cut section, degassed, and cured at 80 °C for 1 h. Finally, the microneedle was removed from the PDMS, leaving behind the channel. The cylindrical channels were created within the PDMS layer. Inlet and outlet holes were then punched at the start and end of the channel using a biopsy punch tool with a 1.5 mm diameter. Following this, the microchannel layer was bonded to a coverslip, and the top layer was then bonded to the microchannel layer, resulting in the formation of the vessel chips. To secure the integrity of the two microchannel ends, a very small amount of PDMS was added to each end of the channel and baked for sealing purposes. The 3D microvessels were sterilized with UV treatment for 20 min and coated with human placenta-derived collagen type IV (400 μg/mL; Sigma) and human plasma-derived fibronectin (100 μg/mL; Sigma) by adding 100 μL of ECM solution into the inlet and outlet holes and incubating at 4 °C overnight.

### 2.3. Cell Seeding in Vessel-on-Chips

The chips were sterilized via UV treatment, loaded with EC medium in a biosafety cabinet, and left to incubate for approximately 30 min. The iBMEC-like cells (at day 6 of iBMEC culture) were then seeded into the microchannel at a concentration of approximately 10 million cells per mL. EC medium was then slowly added to the reservoir on each side. The chips were then left overnight to allow the cell to attach to the channels. On the second day, after preparing devices (day 7 of iBMEC culture), unattached cells were washed away from the chips using fresh medium. Subsequently, the medium was replaced with EC medium without bFGF and RA for further culturing.

### 2.4. Cell Seeding on ECM-Coated PDMS Plates

PDMS was prepared by using a 10:1 ratio by mass of Sylgard 184 Silicone Elastomer Base to 184 Silicone Elastomer Base Curing Agent. The PDMS was degassed in a vacuum desiccator for 20 min to eliminate air bubbles. Approximately 100 μL of the PDMS components were added to each well of the 24-well glass bottom plates (MatTek, Ashland, MA, USA) with gentle shaking to ensure even coverage. The PDMS-coated plates were placed in an 80 °C oven for approximately 1 h. To sterilize the plates, they were exposed to UV light for 20 min. Then, the plates were coated with human placenta-derived collagen type IV (400 μg/mL; Sigma) and human plasma-derived fibronectin (100 μg/mL; Sigma) and incubated at 4 °C overnight. On the following day (day 6 of iBMEC culture), 6 × 10^4^ cells were seeded in each well. The cells were fixed and subjected to immunostaining on day 8 (2 days after seeding onto the PDMS).

### 2.5. TNF-α Treatment in 3D Vessels

Once the iBMEC-like cells formed the vessel in chips (day 8 of iBMEC culture; 2 days after seeding into the 3D vessels), TNF-α (Sigma) was applied to the chips at the concentration of 25 ng/mL in EC medium without bFGF and RA. The 3D vessels were cultured with the TNF-α for 24 h, after which the chips were fixed and used for subsequent immunostaining.

### 2.6. Immunostaining

The iBMEC-like cells in the channel were fixed in 4% paraformaldehyde (Thermo Fisher Scientific) for 10 min at room temperature and then permeabilized for 10 min in 0.25% Triton-X (Sigma) at room temperature. Cells were blocked for nonspecific binding in 3% goat serum (Abcam, Cambridge, UK) for 1 h at room temperature. Samples were then incubated overnight in primary antibodies at 4 °C. The next day, cells were incubated for 1 h with secondary antibody. This step was followed by incubation with 1:1000 Hoechst solution (Thermo Fisher Scientific) for 5 min at room temperature. Fluoromount (Thermo Fisher Scientific) was added to the channel to preserve the fluorescent signal. A detailed list of antibodies is shown in Appendix A.

### 2.7. Confocal Microscopy

Following the cell immunostaining procedure, the BBB vessel-on-chips were imaged using Zeiss LSM980 Airyscan2 and Olympus FV3000 laser scanning confocal microscopes. Zen Blue software 3.3 and ImageJ 1.53f were used for the image processing. Once the channel was visible inside the microscope, z-stack images were taken using a 63×/1.4 NA oil immersion lens with a z-stack spacing of 1.0 µm. A total of 75 z-stacks were obtained for each fluorescence channel on the Zeiss LSM980 Airyscan2. For the FV3000 laser scanning confocal microscopes, the acquisition of z-stack images was performed using a 20× lens with a z-stack spacing of 0.5 µm. Visualization of cell monolayers on the PDMS plates (2D surfaces) was conducted using an Olympus IX83 inverted microscope.

### 2.8. Microchannel Unwrapping

To convert the 3D vessel structure into a 2D surface for cell–cell junction analysis, the z-stack channels were then called into a MATLAB-based image “UNWRAP” program [19]. The channels were then reconstructed by the UNWRAP program and two channel cross-sections were produced for the user to “waypoint”. After waypointing along the circumference of the cross-section, a circle was then fitted to the waypoints to mark the circumference of the channel. Finally, the channel was “unwrapped,” producing a 2D image of the flattened cell monolayer.

### 2.9. JAnaP Analysis

Unwrapped images of cell monolayers from the vessel-on-chip were directly analyzed using our lab’s Junction Analyzer Program (JAnaP, available through https://github.com/StrokaLab/JAnaP, accessed on 25 June 2019) [25]. After loading images to the program, cell boundaries were waypointed manually, with the program connecting each waypoint by following the fluorescent staining of cell–cell junctions, allowing the program to identify the perimeter. Using a Jupyter Notebook, threshold values for fluorescent signals were determined to eliminate noise and best isolate junctions through each data set. Then, the JAnaP program assigned junction phenotype based on path length and thickness-to-path length ratio. A path length greater than 15 pixels (~2.7 µm for a 1024 pixel × 1024 pixel image) was categorized as a continuous junction. Junctions with a thickness to path length ratio greater than 1.2 were counted as perpendicular, while less than 1.2 indicated a punctate junction.

### 2.10. Statistical Analysis

GraphPad Prism 8 was used for all statistical analysis and graph generation. For statistical analysis, the D’Agostino–Pearson normality test was performed to identify the normality of the data. If the data was normal, a *T*-test was used for analysis. Error bars represent the standard deviation of the mean as noted in the figure caption. Statistical significance was indicated as * *p* < 0.05, ** *p* < 0.01, *** *p* < 0.001, and **** *p* < 0.0001.

## 3. Results

### 3.1. Fabrication of Chips in Approximately 3 h

The vessel-on-chips were fabricated according to the protocol described in the Methods section and following the steps shown in Figure 1. Generally, two PDMS layers need to be prepared in this method. For preparing the bottom layer, or a microchannel layer, a rectangular master mold was cut from a polymerized PDMS from a blank silicon wafer (Figure 1a,b). The dimension of 20mm (width) × 25mm (length) was used to fit the chip to a 75 mm × 25 mm coverslip (Figure 2a). The height of the master mold was around 0.3–0.5 cm. Next, a rectangular section was cut in the center of the master mold (Figure 2b). To make uniform the PDMS thickness and imaging distance (Z) from the bottom of the chip to the bottom edge of the microchannel, double-sided tape was used to fix the microneedle on the empty wafer. Meanwhile, two slits were made at both sides of the master mold (Figure 2c) and the device was flipped over to anchor the needle in the center (Figure 1c and Figure 2d). This method generated uniform thickness for the imaging across the length of the microchannels compared to the technique without double-sided tape anchoring.

The PDMS mix was poured into the middle rectangular section and cured (Figure 1e and Figure 2e). After removing the microneedle from the PDMS, the 3D microchannel was obtained across the master PDMS mold. Although the diameter of the microneedle was 200 µm, the diameter of the microchannel generated was measured to be around 160 µm via microscopy. A 1.5 mm diameter puncher was used to make the inlets and outlets to the channel (Figure 2f). A blank PDMS layer was used as the top reservoir for the vessel-on-chip (Figure 2g). To make sure the reservoirs could hold enough medium for the cell culture, the thickness of the top layer was around 0.5–1 cm. Punches of 4–6 mm were used to make reservoirs. Then, the microchannel layer was bonded to a coverslip, and the top layer was bonded to the microchannel layer to form the vessel-on-chips (Figure 1g–i and Figure 2h). This method can produce vessel-on-chips within three hours without using any complicated soft lithography processes. Moreover, this method allows users to customize the chip size and chip number, for example, to bond multiple chips (3–5 chips) on one coverslip to scale up the experimental design, which makes it easier to plan for different treatments and technical replicates (Figure 1h).

### 3.2. Three-Dimensional Vessel-on-Chip Formation and Imaging

The chips were sterilized with UV light and washed with ethanol and PBS. The washing was also performed to test if there was any leaking in the system. After testing and coating the microchannels with ECM protein, the iBMEC-like cells were seeded into the microchannels on day 6 of the iBMEC culture (Figure 1k and Figure 2i). It was important that the cells were dissociated into single cells. The cell clumps blocked the microchannels and failed to attach to the walls of the microchannels. We determined that 10 million cells/mL with a microvessel diameter of 160 µm was optimal for cell seeding and attachment in our system. Lower cell concentrations resulted in the suboptimal covering of the microchannel walls with iBMECs. Meanwhile, vessel diameters lower than 160 µm generated excessively high frictional resistance for the fluid and cells to flow through properly, and higher vessel diameters could not be captured fully by the Airyscan microscope to develop a complete 3D image of the microvessel. After 48 h (day 8 of iBMEC culture), the cells spread and covered the surface of the microchannels, forming the microvessel structures, and the treatments were then applied on day 8 for the following experiments (Figure 1k). As shown in Figure 3a, a full, 3D vessel lined with iBMEC-like cells was successfully generated and imaged using the methods above. A longitudinal cross-section of this vessel at the midline is shown in Figure 3b. The 3D structure of part of the microvessels is shown in Appendix A. The chips were collected, and the cells were immunostained and imaged via Airyscan microscopy as described above.

### 3.3. Three-Dimensional Image Stacks Were Converted to 2D Images for Junction Analysis

After imaging the 3D microvasculature using the Airyscan microscope, the 3D structure can be read by other programs by saving hundreds of longitudinal cross-sections (z-stacks) (Figure 3a,b). We determined that the MATLAB “UNWRAP program” can be used to successfully reconstruct the 3D vessel and “unwrap” it into a 2D sheet [31]. Normally, all the z-stacks would be supplied to the program, but because the program has a maximum number of z-stacks it can receive before exceeding memory, only half of the z-stacks were supplied as the input. Hence, the resulting image was a half-cylinder rather than a full-cylinder. MATLAB displays to the user a cross-section of this half-cylinder (Figure 3c). The user must then “waypoint” along the circumference of the cylinder to instruct MATLAB on the location and curvature of the cylinder (Figure 3d). Once the circumference is waypointed, MATLAB can proceed to unwrap the cylinder to a 2D sheet. We determined that the program successfully unwrapped the half-cylinder z-stacks. As shown in Figure 3d, the program fit a semicircle along the vessel’s circumference, allowing the successful unwrapping of the 3D structure to a 2D surface. The cellular network’s 2D morphology is shown in Figure 3e. After analyzing the 2D unwrapped image in Figure 3e using the JAnaP, we obtained quantitative results characterizing the cellular network using the scheme shown in Figure 4a–e.

### 3.4. Three-Dimensional Structure Altered the Expression and Presentation of the Tight Junction Proteins

Tight junctions play a crucial role in maintaining the blood-brain barrier’s permeability by forming restrictive sealing elements. In this study, we focused on evaluating the specific tight junction protein, ZO-1, to investigate any alterations in junction presentation when the cell monolayer adopted a 3D cylindrical structure. ZO-1 is of particular interest due to its linkage between the actin cytoskeleton and homophilic cell–cell junction proteins, and we hypothesized that the ZO-1 phenotype could depend on morphological changes that occur in cell arrangements in 3D vessels vs. on 2D surfaces. Our findings indicated significant reductions in continuous junctions (Figure 4f) and total ZO-1 coverage (Figure 4g) in 3D vessels compared to cells cultured in 2D on PDMS plates. Additionally, morphological changes were observed, with increased cell area (Figure 4h) and parameter (Figure 4i), along with decreased circularity (Figure 4j) and solidity (Figure 4k) for the cells in the 3D vessels. Furthermore, the expression of Claudin-5 disappeared in the 3D vessels. Collectively, these results demonstrate altered junction expression and integrity in the 3D vessel model compared to the 2D cell culture.

### 3.5. TNF-α Disrupted the Tight Junction Presentation in 3D Vessels

To assess the utility of this system for testing the effects of stimuli on barrier function, we treated the 3D vessels with TNF-α for 24 h and then examined the expression and junction presentation of ZO-1 and Occludin. The results revealed a significant reduction in continuous junctions (Figure 5a) and the total coverage of ZO-1 (Figure 5b) following TNF-α treatment. Moreover, the continuous junctions, perpendicular junctions, and total coverage of Occludin also decreased with TNF-α treatment (Figure 5c,d). Notably, Occludin exhibited higher sensitivity to the TNF-α treatment in the 3D vessels. Additionally, morphological changes were observed (Figure 6), with an increased cell perimeter (Figure 6a) and area (Figure 6b) in the TNF-α-treated cells. These findings demonstrate that TNF-α disrupts the tight junction presentation in the 3D vessels, highlighting the potential of this model for studying barrier function disturbances caused by specific stimuli.

## 4. Discussion

In this study, the microchannel was made by directly polymerizing PDMS around a microneedle. Human iPSC-derived BMEC-like cells were loaded into the ECM-coated engineered microchannels to form the 3D microvessels. By using a MATLAB UNWRAP program, we reconstructed the 3D microvascular structure and unwrapped the 3D image to a 2D surface of the cells. The surface was then quantitatively analyzed by our Python-based JAnaP. Using this method, we were able to obtain quantitative metrics for iBMEC-like cells in a 3D culture, including a quantitative breakdown of the cell–cell junction phenotype presentation, cell morphology, and cell size. These results establish the feasibility of this method for future in vitro studies of the BBB and have the potential to impact the biomedical engineering industry by allowing engineers to streamline drug development testing or develop methods for earlier BBB disease diagnosis.

The BBB plays a key role in maintaining the health of the brain tissue and other central nervous system (CNS) structures [32]. Many studies have utilized 2D models of the BBB and generated informative and useful results. However, the brain microvessels in vivo are 3D structures and the BMECs lining these capillaries exhibit a distinctive shape and function compared to their 2D counterparts. These features have been reviewed in detail elsewhere [33,34]. Moving our in vitro models to 3D will likely have translational benefits and generate results that are closer to the physiological situation. Recently, microfluidics has improved our ability to create 3D in vitro models of the BBB [35,36,37]. Meanwhile, our lab’s custom Python-based JAnaP has already been shown to detect differences in the cell–cell junction phenotypes and barrier integrity in 2D monolayers using a quantitative approach [25,26,27,28,38]. Here, we combined the fabrication of 3D vessels and analysis approaches and developed an in vitro model of the vessel-on-chip that we believe will be very useful in studying the BBB responses to mechanical and chemical cues.

The major advantage of this method is its low cost and short time for fabrication, as well as its simple fabrication process. Different techniques have been used to construct microvascular tube structures, including the insertion of microfibers [39], microneedles [15], glass rods [19], or nitinol wire [40,41] into a gel matrix before polymerization. Moreover, self-organized microvascular networks have been generated to mimic the natural processes of the angiogenesis process [42], where endothelial cells sprout from preexisting vascular channels and self-assemble into branched vessels within adjacent ECM gels [43,44]. Previous methods used, for the channel formation involved, multiple fabrication steps and layers, making them time-consuming, challenging to handle, and requiring specialized skills. In contrast, our model represents a more user-friendly and reproducible approach, with easy-to-follow steps that do not demand specific fabrication skills for the operator. Additionally, it is still challenging to image and analyze the junction expression and junction presentation in the branched vessels. Compared to a 3D-printing or soft-lithography approach [6,45], our protocol allows users to fabricate custom chips for both educational and industrial environments without a 3D printer or fabrication facility. Moreover, our research implements an unwrap technique [19] to expand 3D blood vessels and generate 2D images, which are then analyzed using JAnaP to assess the distribution of the junction. Our data analysis pipeline provides a comprehensive assessment of the cell–cell junction and morphological phenotypes of iBMEC-like cells cultured in 3D microvessels. This analysis is crucial as the distribution of cell–cell junctions is often directly linked to the permeability of blood vessels.

Endothelial functions have been predominantly studied using 2D cell culture models. However, in our current study, we have undertaken a comparative analysis of tight junction marker expression to assess the expression of the tight junction proteins in iBMEC-like cells. Specifically, we have focused on the critical tight junction proteins ZO-1, Claudin-5, and Occludin, which collectively form an intricate network and serve as principal hubs responsible for regulating the physical barrier properties of the blood-brain barrier (BBB). Among these proteins, ZO-1 plays a pivotal role by binding to the actin cytoskeleton, acting as a structural bridge that connects transmembrane proteins with cytoskeletal components. On the other hand, Occludin and Claudin-5 represent key constituents of the tight junction strand in brain endothelial cells, being indispensable for tight junction formation and the precise regulation of BBB permeability [46]. Our investigation has revealed that the 3D geometry significantly impacts the expression of tight junctions and the morphological factors in iBMEC-like cells. These findings strongly suggest that the 3D structure also exerts a regulatory influence on the barrier function of the BBB. As a result, it has become imperative to consider and study the barrier function of the BBB under more physiological geometries to gain a comprehensive understanding of its intricate mechanisms. To ascertain the robustness of our models for evaluating the impact of the perturbance factors on BBB barrier function, we conducted experiments involving TNF-α, a well-known disruptor of BBB function. Conventionally, TNF-α induces a loss in the barrier properties by decreasing the expression of junction proteins, elevating permeability, and reducing the TEER in BMECs [47]. In agreement with these findings, our system exhibited sensitivity to TNF-α treatment, as we observed distinct alterations in ZO-1 and Occludin in iBMEC-like cells in the 3D vessels following exposure to TNF-α. These findings further underscore the reliability and potential of our system for conducting investigations involving various stimuli that may potentially disturb the barrier functions of the BBB.

Evaluating the permeability of the blood vessels in 3D vessels has posed a significant challenge [48]. Traditionally, transendothelial electrical resistance (TEER) is utilized for in vitro barrier function evaluation [49], but this technique does not easily translate to 3D microvessels since the measurement of TEER from the ends of a long channel is challenging and the long electrodes are required in both luminal and abluminal spaces [50]. While some studies have employed the construction of vascular structures within hydrogels, and employed fluorescently labeled small molecules for permeability analysis, the technique is relatively complex and presents technical difficulties for batch screening [51,52,53]. Furthermore, incorporating additional cell types within the gel further increases technical complexities and limits the feasibility of the rapid assessment of vascular permeability. Recently, our laboratory has employed a local permeability assay that establishes a direct quantitative relationship between junction phenotypes and local permeability [27]. In future work, a local permeability assay can be incorporated into our vessel chips and analysis pipeline, enabling the assessment of spatial heterogeneities in blood vessel permeability and their correlated junction presentation in 3D.

By creating a 3D microvessel, and using the JAnaP to characterize cell–cell junction presentation as a function of mechanical and chemical cues, this method will serve as a “quick and easy” way to independently study how environmental conditions affect barrier function, allowing for a better understanding of how BBB diseases arise and propagate. The limitation of this method lies in its incapacity to facilitate the introduction of additional BBB cells or other brain cells for direct interaction with BMEC cells. Furthermore, the accuracy of the cell source poses challenges. For instance, the existing iBMEC differentiation protocol is subject to controversy, as the generated iBMECs may contain epithelial cell types, thus compromising their identity [54,55,56]. Additionally, the human brain tissue is very soft (~1–8 kPa) [57], and the stiffness of PDMS (MPa range) may not be able to accurately replicate the mechanical microenvironment of the BMECs. However, future iterations of this BBB chip could substitute other types of ECM or synthetic gels for PDMS, resulting in the ability to incorporate other BBB cells (e.g., astrocytes and pericyte) and/or better replicate the mechanical properties of the matrix that BMECs are exposed to in vivo. In our study, the solid wall formed by PDMS may produce an altered phenotype because the abluminal factors may not be removed or metabolized in our system. Additionally, the flow rates have been shown to affect the gene expression of cells [58]. The sustained cultivation of the model over an extended period remains arduous, particularly in the presence of flow, necessitating the ongoing optimization of conditions. As researchers continue to advance the generation of various human pluripotent stem cell-derived BBB cells, and engineer innovative BBB models, challenges persist in integrating all the relevant factors, including different BBB cells, brain cells, ECM, and mechanical cues, into a comprehensive BBB model. The complexity and limitations of the various BBB models must be carefully considered in the context of experimental goals. However, it is still worthwhile to evaluate simplified BBB models to determine the minimum factors necessary to include in order to achieve behaviors that are predictive of in vivo outcomes. Here, we presented a 3D-engineered iBMEC-coated microvessel that serves as a simple endothelial vessel model, and the data analysis pipeline provides the foundation for future evaluations of BMECs in 3D under various disease-related conditions.

## Figures and Tables

**Figure 1 bioengineering-10-01080-f001:**
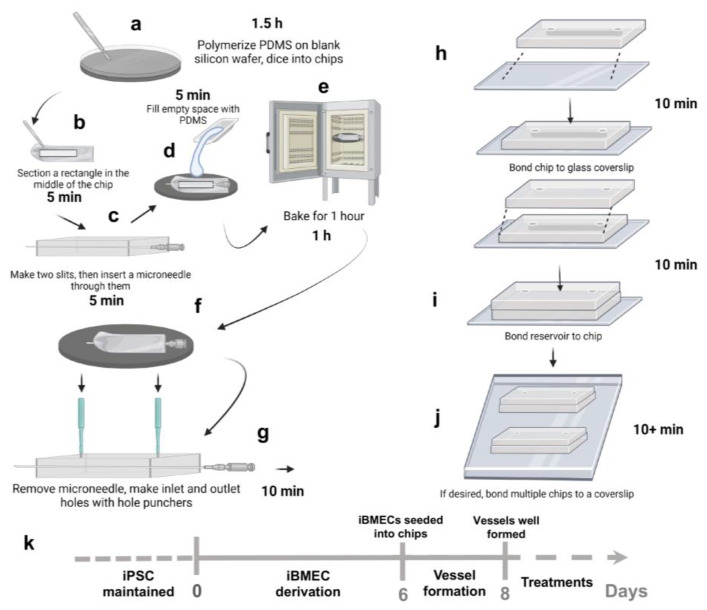
**Schematic overviewing the fabrication process of our in vitro 3D vessel-on-chip model.** (**a**) PDMS is polymerized on a blank silicon wafer and diced into chips. (**b**) A rectangle section is cut in the middle of the chips. (**c**) Two slits are made in the bottom layer and then a microneedle is inserted through it. (**d**) Space-filling with PDMS. (**e**) PDMS device baking. (**f**) Removal of microneedles. (**g**) Punching of inlets and outlets. (**h**) Preparation of top layer and bonding. (**i**) Bonding reservoir to chip (**j**) Multiple chips in one coverslip. (**k**) The iPSCs were cultured on Matrigel-coated 6-well plates and differentiation was initiated on day 0. The iBMEC-like cells were subcultured into the ECM-coated microchannels on day 6. On day 8, vessels were well formed in the channels and treatments were introduced into system. Figure was generated by BioRender.

**Figure 2 bioengineering-10-01080-f002:**
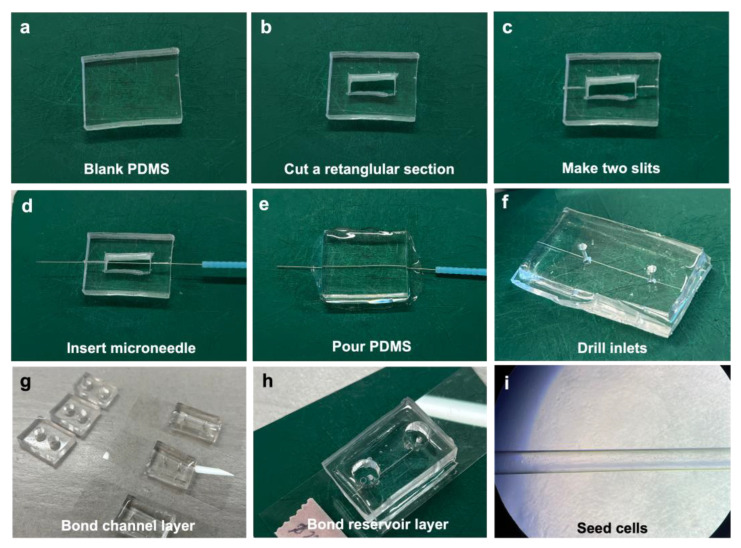
**Overview of fabrication process.** (**a**) For the fabrication process, PDMS is polymerized on top of a blank silicon wafer, diced, and extracted into 20 mm × 25 mm chips. (**b**) A 5 mm × 15 mm section is then cut from the middle to allow the microneedle to pass through. (**c**) Two slits are made along the rectangular section bisection line. (**d**) The microneedle is inserted within the slits. (**e**) The chips are flipped over, microneedle side down, and placed back on the wafer, where PDMS is poured into the cut section and polymerized. (**f**) The microneedle is then pulled and two ø 1.5 mm holes are punched along the channel path to make a channel inlet and outlet. (**g**) The reservoir layers are made by punching two 7 mm holes in blank 20 mm × 25 mm PDMS chips. (**h**) Channel layer and reservoir layer are bonded to a glass coverslip. (**i**) Brightfield image of microchannel with iBMEC-like cells.

**Figure 3 bioengineering-10-01080-f003:**
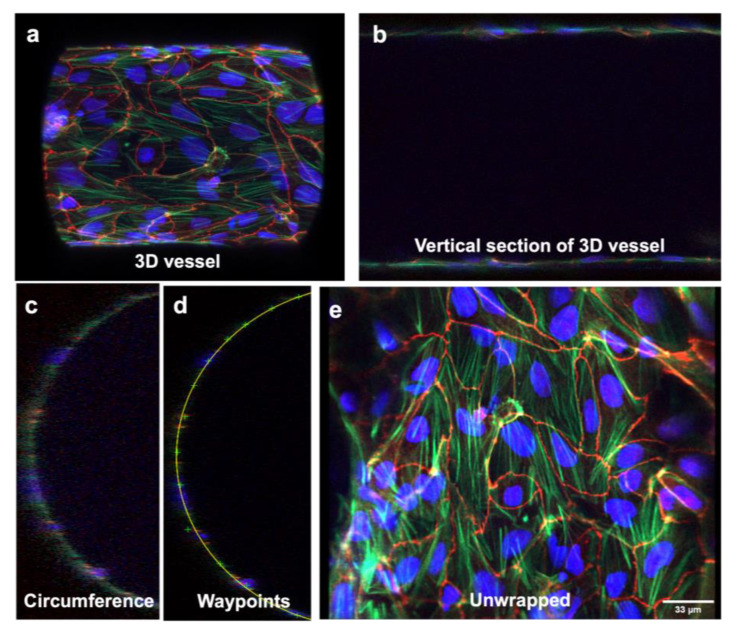
**Unwrapping 3D immunostained image stacks into 2D surfaces.** (**a**) 3D image of BBB channel reconstructed by the UNWRAP program. (**b**) A vertical section of the BBB channel. (**c**) A semicircular cross-section of the BBB channel. (**d**) A circle is then fitted to the channel circumference based on user-specified waypoints along the channel’s path. (**e**) The channel is then “unwrapped” into a 2D surface. Scale bar represents 33 μm.

**Figure 4 bioengineering-10-01080-f004:**
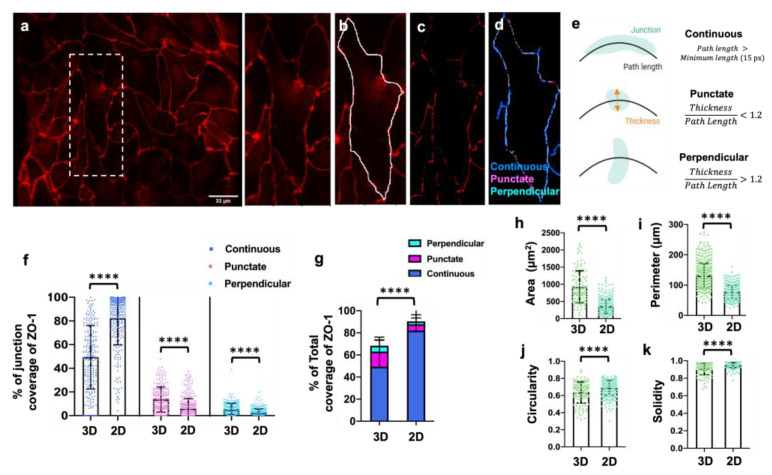
**Analysis of cell–cell junctions and cell morphology in unwrapped images.** (**a**) An unwrapped 2D image. The dotted outline box in the left image is shown zoomed in the right image of this panel. This zoomed image also corresponds to the image in panels (**b**–**d**). (**b**) Cell is identified by the JAnaP when the user “waypoints” a cell along its border. Once all the cells have been waypointed, the JAnaP then processes each cell. (**c**) In a particular cell of interest, the JAnaP will then apply a filter along the user-specified cell border to eliminate background. (**d**) Along the cell border, cell junctions are then classified according to the scheme indicated. (**e**) The classified cell junctions are displayed, and their phenotype data are saved for analysis. (**f**) The presentation of continuous, punctate, and perpendicular junctions for ZO-1 are shown, respectively, for 3D vessel-on-chip devices (3D) and 2D PDMS surfaces (2D). (**g**) The total junction coverage of ZO-1. (**h**–**k**) Cell shape factors based on ZO-1 expression. Note: 203 ≤ N ≤ 297, where N is the number of cells pooled from three biological replicates. In dot plots, each dot represents the value for one cell. Bars represent mean and error bars represent standard deviation. Statistical significance was indicated as **** *p* < 0.0001.

**Figure 5 bioengineering-10-01080-f005:**
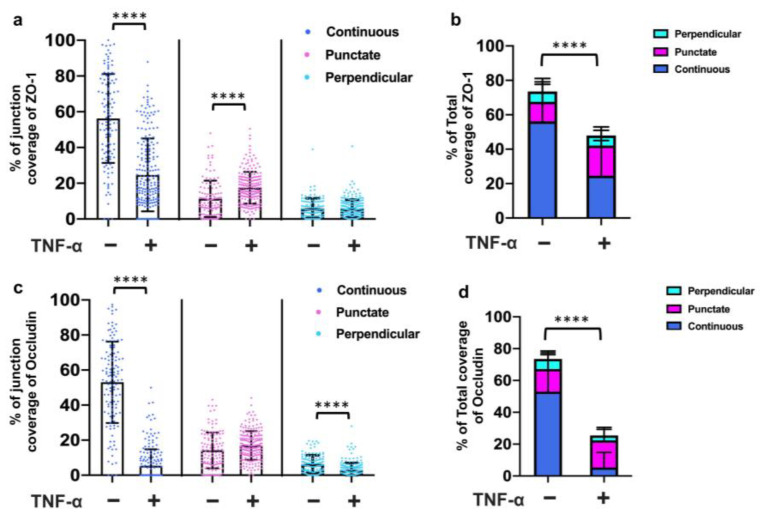
Analysis of cell–cell junctions in 3D vessels with TNF-α treatment. (**a**) The presentation of continuous, punctate, and perpendicular junctions for ZO-1 are shown, respectively. (**b**) The total junction coverage of ZO-1. (**c**) The presentation of continuous, punctate, and perpendicular junctions for Occludin are shown, respectively. (**d**) The total junction coverage of Occludin. Note: 119 ≤ N ≤ 211, where N is the number of cells pooled from three biological replicates. In dot plots, each dot represents the value for one cell. Bars represent mean and error bars represent standard deviation. Statistical significance was indicated as **** *p* < 0.0001.

**Figure 6 bioengineering-10-01080-f006:**
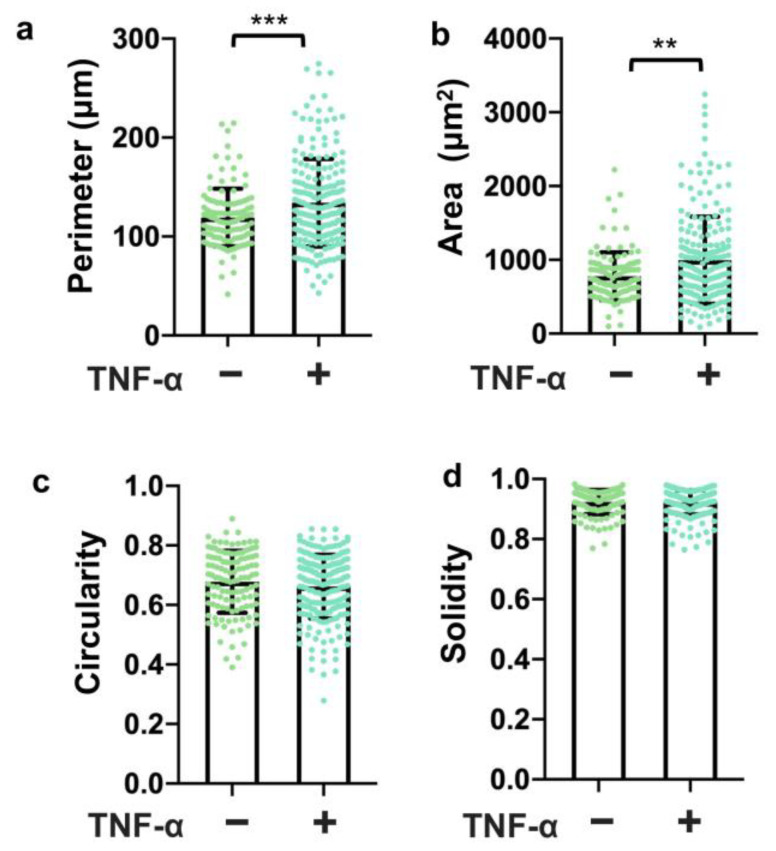
Analysis of cell morphology in 3D vessels with TNF-α treatment. Morphologies analyzed include (**a**) perimeter, (**b**) area, (**c**) circularity, and (**d**) solidity. Note: 119 ≤ N ≤ 211, where N is the number of cells pooled from three biological replicates. In dot plots, each dot represents the value for one cell. Bars represent mean and error bars represent standard deviation. Statistical significance was indicated as ** *p* < 0.01, *** *p* < 0.001.

## Data Availability

The JAnaP is available for download at https://github.com/StrokaLab/JAnaP (accessed on 25 June 2019).

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
