# Peer review of "A Rapid-Patterning 3D Vessel-on-Chip for Imaging and Quantitatively Analyzing Cell–Cell Junction Phenotypes"

_bioengineering, 2023, doi:10.3390/bioengineering10091080_

Round 1
Reviewer 1 Report
This paper presents a method for growing and imaging endothelial cells inside of a circular PDMS channel. The title of manuscript is misleading and does not reflect the actual content, particularly since the walls of the PDMS channel are impermeable and do no support other key cell types outside of the endothelial cells that modulate the blood-brain barrier permeability, including pericytes and astrocyes. The paper is best described by the author’s own statement from line 90: “a proof-of-concept 3D microvessel model,” and it is a microvessel model for which barrier penetration cannot be determined since there is no abluminal space in which transported compound could accumulate. Yes, it is possible to get high quality confocal images of the cells that can be analyzed with an unwrapping algorithm, but that is about all that this assay was shown to do. No particularly interesting biology is demonstrated.
Neither the methods or the results are particularly novel, and there are much earlier reports in the literature of simple-to-use, now-commercially available chips that allow the casting of circular channels in a hydrogel, for which the abluminal space is both accessible and can support multiple other cell types. See for example “Brief Communication: Tissue-engineered Microenvironment Systems for Modeling Human Vasculature,” Anna Tourovskaia, Mark Fauver, Gregory Kramer, Sara Simonson, and Thomas Neumann, Exp Biol Med (Maywood). 2014 September; 239(9): 1264–1271. doi:10.1177/1535370214539228 and the product information from Nortis, Inc., and more recent papers by others.
The novelty of the presented research is not clear. If the goal is to establish fast and reliable approach for microchannel fabrication then introduction needs to be expanded to cover other fabrication approaches / techniques and then compare those approaches to the one presented here. What are the limitations of the presented approach to channel molding? Who else has pulled wires from PDMS to create circular channels? How do authors imagine fabrication of branched network using microneedles and or thin wires?
The use of of only brain microvascular cells is an extreme overstatement the barrier is not comprised of a single cell type and the lack of any other cell types is a concern as well as having zero access to the supposed brain space. The argument that a monoculture of induced BMEC-like cells constitutes a model of the blood-brain barrier is overreach. The novelty of the very conventional circular microfluidic channel is overstated. It is unclear how their channel method could be functionally expanded to include glial cells and make the model more biologically relevant. The one claim to novelty that this manuscript can make is the coupling of their JAnaP software to the UNWRAP software developed in another lab, and this is a pretty thin claim, given that they don't use the coupling to show any interesting biology.
Relevance of a single channel molded in solid material with a single cell type to the formation of BBB and changes in cell permeability is not evident. Also discussions of TEER and other measurements of vessel permeability presented in Discussion section” on lines 357-369 in the context of presented approach are irrelevant. Without a central wire down the channel, measurement of TEER from the ends of a long channel is highly inaccurate, and without access to the abluminal space, impossible. See for example Odijk, et al., “Measuring direct current trans-epithelial electrical resistance in organ-on-a-chip microsystems,” Lab Chip, 2015, 15: 745 DOI: 10.1039/c4lc01219d.
The absence of a perfused abluminal space can also lead to physiologically unrealistic gene expression, as shown by Shin et al., iScience 15, 391 – 406, May 31, 2019 ª 2019 https://doi.org/10.1016/, j.isci.2019.04.037
Details of culture conditions for BMECs within the channel are missing. Was it a static culture or was there a flow present? If flow, what were the shear forces to which the cells were exposed? If it is static culture then then how relevant the “tight junction staining.” Have authors done culture under the flow conditions? Was there a change in cell morphology and formation of tight junctions in presence of shear forces?
Minor comments:
Line 44 - animal models are accurate, but often lack relevance to humans
Line 86 - missing reference
Line 94 - references made to nitinol wires are not followed-through with in the methods, where stainless steel accupuncture needles are described. The authors need to be clear about which (or if both) were used in the experiments described in the manuscript.
Line 155 - no discussion of what the ZO protein is or why it was selected is made. This is of crucial importance to the goal of the paper, so should be explained in some detail.
Line 187 xxx ?
Line 247 - "microchannel ends were sealed with vacuum grease". Arguments about the cell-compatibility of PDMS aside, a better method than vacuum grease should be tried here. A third baking step using PDMS to seal the ends of the microchannel would be better.
Line 265 - "it is not recommended to maintain a staic culture of iNMEC-like cells more than Day 10" If you have a microchannel, why not establish flow and prolong the experiment to a more useful duration? Flow will also aid in the formation of tight junctions.
Line 273. Missing citation.
Line 291. Four significant digits with an uncertainty of 22% is meaningless.
In order to publish this, they should either:
1) show more interesting biology with their device/pipeline, to establish that it is worth having and is better than commercial alternatives, and narrow the scope of their manuscript to vascular components of the BBB.
OR
2) - Apply the UNWRAP/JAnaP combination to a more complete BBB model, i.e., with a astrocyte/pericyte component.
Author Response
Comments from reviewer 1
Reviewer comments: This paper presents a method for growing and imaging endothelial cells inside of a circular PDMS channel. The title of manuscript is misleading and does not reflect the actual content, particularly since the walls of the PDMS channel are impermeable and do no support other key cell types outside of the endothelial cells that modulate the blood-brain barrier permeability, including pericytes and astrocyes. The paper is best described by the author’s own statement from line 90: “a proof-of-concept 3D microvessel model,” and it is a microvessel model for which barrier penetration cannot be determined since there is no abluminal space in which transported compound could accumulate. Yes, it is possible to get high quality confocal images of the cells that can be analyzed with an unwrapping algorithm, but that is about all that this assay was shown to do. No particularly interesting biology is demonstrated.
Author response: We thank the reviewer for their comments. We have changed the title to better reflect the content of the manuscript: “A rapid-patterning 3D vessel-on-chip for imaging and quantitatively analyzing cell-cell junction phenotypes.” Furthermore, we have added data from multiple new experiments that demonstrate interesting biology that can be derived from our the model system.
Reviewer comments: Neither the methods or the results are particularly novel, and there are much earlier reports in the literature of simple-to-use, now-commercially available chips that allow the casting of circular channels in a hydrogel, for which the abluminal space is both accessible and can support multiple other cell types. See for example “Brief Communication: Tissue-engineered Microenvironment Systems for Modeling Human Vasculature,” Anna Tourovskaia, Mark Fauver, Gregory Kramer, Sara Simonson, and Thomas Neumann, Exp Biol Med (Maywood). 2014 September; 239(9): 1264–1271. doi:10.1177/1535370214539228 and the product information from Nortis, Inc., and more recent papers by others.
Author response: Indeed, a major goal of this manuscript was to highlight the image analysis pipeline that can be used to take 3D confocal images, unwrap them into 2D surfaces, and then perform quantitative junction analysis. We believe this process will be quite useful for researchers studying cell-cell junction architecture in 3D vessels. We have discussed previous strategies for making circular channels in line 768-774. “Different techniques have been used to construct microvascular tube structures, including insertion of microfibers[39], microneedles [15], glass rods [19], or nitinol wire [40,41] into gel matrix before polymerization. Moreover, self-organized microvascular networks have been generated to mimic the natural processes of angiogenesis process [42] where endothelial cells sprout from preexisting vascular channels and self-assemble into branched vessels within adjacent ECM gels [43,44].”
Reviewer comments: The novelty of the presented research is not clear. If the goal is to establish fast and reliable approach for microchannel fabrication, then introduction needs to be expanded to cover other fabrication approaches / techniques and then compare those approaches to the one presented here. What are the limitations of the presented approach to channel molding? Who else has pulled wires from PDMS to create circular channels? How do authors imagine fabrication of branched network using microneedles and or thin wires?
Author response: In the revised manuscript, we have discussed more limitations of the presented approach for making a vessel-on-chip. We also added more discussion of our methods and other published methods. In our study, we do not focus on the branched network since it is difficulty to imaging and analyze the junction presentation in a branched network. Lines 768-779: “Different techniques have been used to construct microvascular tube structures, including insertion of microfibers[39], microneedles [15], glass rods [19], or nitinol wire [40,41] into gel matrix before polymerization. Moreover, self-organized microvascular networks have been generated to mimic the natural processes of angiogenesis process [42] where endothelial cells sprout from preexisting vascular channels and self-assemble into branched vessels within adjacent ECM gels [43,44]. Previous methods used for channel formation involved multiple fabrication steps and layers making then time-consuming, challenging to handle, and requiring specialized skills. In contrast, our model represents a more user-friendly and reproducible approach, with easy-to-follow steps that do not demand specific fabrication skills for the operator. Additionally, it is still challenging to imaging and analyze the junction expression and junction presentation in the branched vessels.”
Reviewer comments: The use of of only brain microvascular cells is an extreme overstatement the barrier is not comprised of a single cell type and the lack of any other cell types is a concern as well as having zero access to the supposed brain space. The argument that a monoculture of induced BMEC-like cells constitutes a model of the blood-brain barrier is overreach. The novelty of the very conventional circular microfluidic channel is overstated. It is unclear how their channel method could be functionally expanded to include glial cells and make the model more biologically relevant. The one claim to novelty that this manuscript can make is the coupling of their JAnaP software to the UNWRAP software developed in another lab, and this is a pretty thin claim, given that they don't use the coupling to show any interesting biology.
Author response: We appreciate the reviewer’s comment. We agree that the single cell type in our study does not recapitulate the cerebrovascular microenvironment of BBB. There are certainly complicated factors that need to be considered in this system, such as specific cell types (BBB cells and other brain cell types), ECM, cell spatial arrangement, flow etc. Our goal was not to combine all these components into the vessel device, and hence we have changed language in the manuscript away from presenting it as a “BBB-on-a-chip.” Instead, we chose to present the system as a simple vessel-on-chip, highlight the ability to quantitatively assess junction phenotypes in the 3D vessels, and use the device to quantitatively evaluate cell-cell junctions in 2D vs. 3D (Figure 4) and also in 3D in the presence of TNF-alpha (Figure 5). Also, we added a brief discussion of the complexities of fully modeling the BBB: Lines 860-865: “As researchers continue to advance the generation of various human pluripotent stem cell-derived BBB ells and engineer innovative BBB models, challenges persist in integrating all relevant factors, including different BBB cells, brain cells, ECM, and mechanical cues, into a comprehensive BBB model. Complexity and limitations of various BBB models must be carefully considered in the context of experimental goals.”
Reviewer comments: Relevance of a single channel molded in solid material with a single cell type to the formation of BBB and changes in cell permeability is not evident. Also discussions of TEER and other measurements of vessel permeability presented in Discussion section” on lines 357-369 in the context of presented approach are irrelevant. Without a central wire down the channel, measurement of TEER from the ends of a long channel is highly inaccurate, and without access to the abluminal space, impossible. See for example Odijk, et al., “Measuring direct current trans-epithelial electrical resistance in organ-on-a-chip microsystems,” Lab Chip, 2015, 15: 745 DOI: 10.1039/c4lc01219d.
Author response: We agree that the TEER measurements in the long channel would be impossible without access to the abluminal space. We believe this device will be quite conducive to a local permeability assay, which our lab has published in 2D monolayers (Gray et al., Fluids and Barriers of the CNS, 2020; and Yan et al., Acta Biomaterialia, 2023). Future work will integrate this local permeability assay into the 3D vessel-on-chip system, and this method will have great benefits over TEER, including the ability to measure spatial heterogeneities in permeability and barrier function. We have included this possibility in the manuscript discussion (lines 834-842). We discussed the limitation of TEER in lines 826-829: “Traditionally, trans-endothelial electrical resistance (TEER) is utilized for in vitro barrier function evaluation[48], but this technique does not easily translate to 3D microvessels, since the measurement of TEER from the ends of a long channel is impossible without access to the abluminal space[49].”
Reviewer comments: The absence of a perfused abluminal space can also lead to physiologically unrealistic gene expression, as shown by Shin et al., iScience 15, 391 – 406, May 31, 2019 ª 2019 https://doi.org/10.1016/, j.isci.2019.04.037
Author response: This is a great point. We are working on the perfused system to evaluate the effects of shear stress in this model. However, in our results, our system can be used as simple system to quickly screen or test some treatments which may be further evaluated in a more complex BBB system.
Reviewer comments: Details of culture conditions for BMECs within the channel are missing. Was it a static culture or was there a flow present? If flow, what were the shear forces to which the cells were exposed? If it is static culture then then how relevant the “tight junction staining.” Have authors done culture under the flow conditions? Was there a change in cell morphology and formation of tight junctions in presence of shear forces?
Author response: The experiments were performed under static conditions. It is possible to connect this system to a syringe pump or a peristaltic pump, and we are working on integrating that into the microfluidic system. However, in this manuscript, the focus was not on the biological effects of shear stress, but instead on the feasibility of imaging the cell-cell junctions and quantifying the junction presentation in the 3D microvessels.
Minor comments:
Reviewer comments: Line 44 - animal models are accurate, but often lack relevance to humans
Author response: We have added this discussion in lines 187-188: “While in vivo models are accurate at mimicking the BBB environment, they involve a heavy use of animals and often lack relevance to humans[7].”
Reviewer comments: Line 86 - missing reference
Author response: We have added the reference.
Reviewer comments: Line 94 - references made to nitinol wires are not followed-through with in the methods, where stainless steel accupuncture needles are described. The authors need to be clear about which (or if both) were used in the experiments described in the manuscript.
Author response: We have clarified this point. Lines 242-243: “…protocol that creates 3D microvessels with polydimethylsiloxane and stainless-steel acupuncture needles.”
Reviewer comments: Line 155 - no discussion of what the ZO protein is or why it was selected is made. This is of crucial importance to the goal of the paper, so should be explained in some detail.
Author response: We have added discussion about the tight junction proteins in Results and also Discussion. Lines 634-: “Tight junctions play a crucial role in maintaining the blood-brain barrier's permeability by forming restrictive sealing elements. In this study, we focused on evaluating the specific tight junction protein, ZO-1, to investigate any alterations in junction presentation when the cell monolayer adopted a 3D cylindrical structure. ZO-1 is of particular interest due to its linkage between the actin cytoskeleton and homophilic cell-cell junction proteins, and we hypothesized that ZO-1 phenotype could depend on morphological changes that occur in cell arrangements in 3D vesels vs. on 2D surfaces.”
Reviewer comments: Line 187 xxx ?
Author response: We have added the appropriate reference.
Reviewer comments: Line 247 - "microchannel ends were sealed with vacuum grease". Arguments about the cell-compatibility of PDMS aside, a better method than vacuum grease should be tried here. A third baking step using PDMS to seal the ends of the microchannel would be better.
Author response: We thank the reviewer for this great suggestion. For the new experiments in the revised manuscript, we have taken this suggestion and added an additional baking step in our protocol. Line 229-230: “To secure the integrity of the two microchannel ends, an additional baking step employing PDMS was performed for sealing purposes.”
Reviewer comments: Line 265 - "it is not recommended to maintain a static culture of iBMEC-like cells more than Day 10" If you have a microchannel, why not establish flow and prolong the experiment to a more useful duration? Flow will also aid in the formation of tight junctions.
Author response: We have deleted this limitation.
Reviewer comments: Line 273. Missing citation.
Author response: We have added the appropriate citation.
Reviewer comments: Line 291. Four significant digits with an uncertainty of 22% is meaningless.
Author response: We have completely rewritten this section and added more results and figures (Figure 4, 5, and 6).
Reviewer comments: In order to publish this, they should either:
1) show more interesting biology with their device/pipeline, to establish that it is worth having and is better than commercial alternatives, and narrow the scope of their manuscript to vascular components of the BBB. OR 2) - Apply the UNWRAP/JAnaP combination to a more complete BBB model, i.e., with a astrocyte/pericyte component.
Author response: We thank the reviewer for encouraging us to significantly strengthen our manuscript. We have narrowed the scope of our manuscript to a vessel-on-chip and performed new experiments to quantitatively show how tight junction phenotypes compare between 2D and 3D vessel structures and how a biologically-relevant cue, TNF-α, affects junction phenotypes in 3D vessels in vitro.
Reviewer 2 Report
The authors here provide a useful analysis technique of “unwrapping” a 3D stack of a cylindrical vessel that will be useful in assessing cell-cell junctional molecules from 3D stacks. They also demonstrate a simple method for cylindrical channel fabrication. This method could be useful to the field for accurate quantification and assessment of junctional molecules from 3D systems. However, there are a number of experiments missing that are needed to demonstrate the utility of this method: 1) More than just one junctional molecule should be tested, 2) more than one condition should be tested (e.g. compromised junctional expression vs baseline) and 3) 3D should be compared to 2D. As described below this would make the paper much more useful to the field, and without these experiments there are a lot of open questions as to the robustness of this approach.
Additionally the paper claims that this is a BBB model but does not assess any BBB characteristics or have any other cell types present other than “brain microvascular endothelial (iBMEC)-like cells”. To derive these cells they use a method by Hollmann, et al. however such methods have been put into question regarding the endothelial identity https://doi.org/10.1073/pnas.2016950118 , presumably this is why the authors refer to them as "iBMEC-Like cells" rather than "iBMEC". The authors should make this more clear and discuss as a limitation to this study. I would suggest that unless the authors provide further characterisation of the BBB phenotype in their model that they change the title of their paper to omit the term BBB. This is not adequately characterised as a BBB on chip. But could be called a vessel on chip.
As mentioned above, only one junctional molecule was tested for ZO-1. This is not a brain specific junctional molecule, further supporting that this model should not be called a BBB model. Also the authors should verify their model and analysis process with another junctional molecule if they are to state that this is a method for “quantitatively analyzing cell-cell junction phenotypes”. In my hands ZO-1 stains very clearly to the cell-cell junction, whereas many other junctional molecules stain in a less well defined pattern, and may present more of a challenge. The authors have shown the protocol to work with a relatively simple example, they should also demonstrate that it will work with another example.
With regard to the need to compare two different conditions in their model: The authors claim this is a method to analyse cell-cell junction phenotypes yet no other phenotype is presented other than the baseline condition. So perturbation of tight junctions is required to make a comparison and prove that the system is useful in comparing phenotypes and is sensitive enough to detect relevant changes. This perturbation could simply be an inflammatory challenge, or a disease specific challenge, like hypoxia for stroke. But it is essential that the protocol test compare two different conditions to demonstrate that it is useful for future research into BBB disruption or cell-cell junction phenotypes.
With regard to 2D vs 3D: The authors also state the importance of working in 3D vessels and that this affects junctional arrangement/cell shape. It would be very useful if the authors could make a comparison between a 2D and 3D cell monolayer using their analysis technique.
One minor point that would also improve the paper: The authors refer to day numbers in the protocol. A diagram showing the key processes and time points should be added to help orientate the reader to the workflow.
Author Response
Comments from reviewer 2
Reviewer comments: The authors here provide a useful analysis technique of “unwrapping” a 3D stack of a cylindrical vessel that will be useful in assessing cell-cell junctional molecules from 3D stacks. They also demonstrate a simple method for cylindrical channel fabrication. This method could be useful to the field for accurate quantification and assessment of junctional molecules from 3D systems. However, there are a number of experiments missing that are needed to demonstrate the utility of this method: 1) More than just one junctional molecule should be tested, 2) more than one condition should be tested (e.g. compromised junctional expression vs baseline) and 3) 3D should be compared to 2D. As described below this would make the paper much more useful to the field, and without these experiments there are a lot of open questions as to the robustness of this approach.
Author response: We thank the reviewer for noting the potential usefulness of our device and analysis pipeline. We have followed the reviewer’s suggestion and completed all three experiments that the reviewer noted as missing. We appreciate the reviewer’s comments and the opportunity to strengthen our manuscript with these new data.
Additionally the paper claims that this is a BBB model but does not assess any BBB characteristics or have any other cell types present other than “brain microvascular endothelial (iBMEC)-like cells”. To derive these cells they use a method by Hollmann, et al. however such methods have been put into question regarding the endothelial identity https://doi.org/10.1073/pnas.2016950118 , presumably this is why the authors refer to them as "iBMEC-Like cells" rather than "iBMEC". The authors should make this more clear and discuss as a limitation to this study. I would suggest that unless the authors provide further characterization of the BBB phenotype in their model that they change the title of their paper to omit the term BBB. This is not adequately characterized as a BBB on chip. But could be called a vessel on chip.
Author response: We appreciate this comment and completely agree. We have changed the title of our manuscript, changing “BBB-on-chip” to “vessel-on-chip.” We also narrowed the scope of the manuscript to focus on a vessel-on-chip rather than emphasizing BBB-on-chip. We have added discussion of the iBMEC-like cells in our manuscript in lines 853-855: “Furthermore, the accuracy of cell source poses challenges. For instance, the existing iBMEC differentiation protocol is subject to controversy, as the generated iBMECs may contain epithelial cell types, thus compromising their identity[53-55].”
Reviewer comments: As mentioned above, only one junctional molecule was tested for ZO-1. This is not a brain specific junctional molecule, further supporting that this model should not be called a BBB model. Also the authors should verify their model and analysis process with another junctional molecule if they are to state that this is a method for “quantitatively analyzing cell-cell junction phenotypes”. In my hands ZO-1 stains very clearly to the cell-cell junction, whereas many other junctional molecules stain in a less well defined pattern, and may present more of a challenge. The authors have shown the protocol to work with a relatively simple example, they should also demonstrate that it will work with another example.
Author response: We thank the reviewer for this excellent suggestion. In the revised manuscript, we have added results for Occludin and Claudin-5. These new results are included in Sections 3.4 and 3.5 in the manuscript. Indeed, Claudin-5 disappeared from iBMEC-like cells in the 3D vessels, while Occludin was present (Figure 5).
Reviewer comments: With regard to the need to compare two different conditions in their model: The authors claim this is a method to analyze cell-cell junction phenotypes yet no other phenotype is presented other than the baseline condition. So perturbation of tight junctions is required to make a comparison and prove that the system is useful in comparing phenotypes and is sensitive enough to detect relevant changes. This perturbation could simply be an inflammatory challenge, or a disease specific challenge, like hypoxia for stroke. But it is essential that the protocol test compare two different conditions to demonstrate that it is useful for future research into BBB disruption or cell-cell junction phenotypes.
Author response: We thank the reviewer for this excellent suggestion. In new experiments, we added TNF-α into the 3D vessel and confirmed that this model can be used for evaluating changes in the barrier integrity under inflammatory conditions (Section 3.5 and Figure 5).
Reviewer comments: With regard to 2D vs 3D: The authors also state the importance of working in 3D vessels and that this affects junctional arrangement/cell shape. It would be very useful if the authors could make a comparison between a 2D and 3D cell monolayer using their analysis technique.
Author response: Again, we thank the reviewer for this great suggestion. We have added the results of the comparation between the 2D and 3D cells in Section 3.4 and Figure 4. Interestingly, we found significant differences between the ZO-1 junction phenotypes in 2D vs. 3D, with less continuous and more discontinuous junctions in the 3D vessels.
Reviewer comments: One minor point that would also improve the paper: The authors refer to day numbers in the protocol. A diagram showing the key processes and time points should be added to help orientate the reader to the workflow.
Author response: We have made a few clarifications in the manuscript. In the Methods Section 2.1, we added: “Day 0 indicates the time of seeding cells on the Matrigel-coated surface.” Then, each time we note the day number, we clarify that it is “of iBMEC culture” and also clarify days after seeding into the vessel-on-chips. We believe that these clarifications will help readers understand the timing of the process.
Reviewer 3 Report
The paper by Dwiggins and colleagues describes a new procedure to obtain a simple model of the 3D microvasculature of the BBB, overcoming some limitations regarding the ability to monitor and evaluate junctional proteins’ expression through microscopy that current alternative methods may present. It describes the methodological aspects of the model development and also stresses the importance of the new analytical methods employed to retrieve a semiquantitative assessment of ZO-1 expression/organization, a clear asset for future studies. It is a simple report, though some aspects should be clarified and/or corrected before being suitable for publication.
Major aspects:
- There are already several protocols published of BBB models based on PDMS microfabrication, some of them even incorporating microfluidics and medium shear stress, closer to physiological that allow longer culture times and to incorporate other neovascular unit cells, such as astrocytes and pericytes. Would it be possible at least to upgrade this model to be coupled with microfluidics? If so, which differences would be expected?
- How many PDMS microfabrication and cell seeding/BBB differentiation protocols were performed? Is the method reproducible from batch to batch? And what about the technical skills of the operator?
- Figure 4: regarding panels F and G, are there any significant differences among the different parameters evaluated? How many biological replicates were done? If so, please include the respective significances and statistical analysis in the methods section and discuss all these relevant aspects in the manuscript.
- The junctional analysis evaluation by the new analytical method developed clearly presents advantages in providing a semi-automated quantitative evaluation of cell junction morphology, which reflects the barrier health/integrity/tightness among possible different treatments. In this regard, did the authors test compounds known to induce BBB leakage and/or to increase BBB tightness to support the usefulness of the model implemented? If so, it should be included; if not, it should at least be discussed the anticipated results in the main text. Moreover, could this method of analysis be applied/transposed to other BBB models besides the one described in this manuscript? And to other TJs and/or AJs proteins? Please discuss.
General aspects/typos:
Authors’ email – different fonts, different from the journal formatting. Please correct.
Main text – different formatting than the one required by the journal, with extra spacing. The same is true for figure captions. Please correct.
Grammar – consider revising English language before publication
Line 86 – missing ref
Material and methods – commercial brands of some used reagents are missing (e.g. PDMS, UCT, etc). Please include.
Line 138 – RA states for ascorbic acid? Or retinoic acid? Please revise.
Figure 4 – Figure caption only goes until E while the figure panels go until letter I. Please complete figure legend.
Consider revising English language before publication
Author Response
Comments from reviewer 3
Reviewer comments: The paper by Dwiggins and colleagues describes a new procedure to obtain a simple model of the 3D microvasculature of the BBB, overcoming some limitations regarding the ability to monitor and evaluate junctional proteins’ expression through microscopy that current alternative methods may present. It describes the methodological aspects of the model development and also stresses the importance of the new analytical methods employed to retrieve a semiquantitative assessment of ZO-1 expression/organization, a clear asset for future studies. It is a simple report, though some aspects should be clarified and/or corrected before being suitable for publication.
Major aspects:
Reviewer comments: There are already several protocols published of BBB models based on PDMS microfabrication, some of them even incorporating microfluidics and medium shear stress, closer to physiological that allow longer culture times and to incorporate other neovascular unit cells, such as astrocytes and pericytes. Would it be possible at least to upgrade this model to be coupled with microfluidics? If so, which differences would be expected?
Author response: It is possible to connect this system to a syringe pump or a peristaltic pump, and we are working on integrating that into the microfluidic system. However, in this manuscript, the focus was not on the biological effects of shear stress, but instead on the feasibility of imaging the cell-cell junctions and quantifying the junction presentation in the 3D microvessels.
Reviewer comments: How many PDMS microfabrication and cell seeding/BBB differentiation protocols were performed? Is the method reproducible from batch to batch? And what about the technical skills of the operator?
Author response: This is very easy PDMS fabrication process, which is why we emphasize it as being “easy,” “simple,” and “fast.” Two undergraduate students in our lab quickly picked up the technique, and the second author (Cole Dwiggins) was an undergraduate student when he completed the work for this manuscript. We added (lines 779-781): “…In contrast, our model represents a more user-friendly and reproducible approach, with easy-to-follow steps that do not demand specific fabrication skills for the operator.” Furthermore, the method is reproducible from batch to batch, producing similar trends in data across 3 independent device trials for each condition.
Reviewer comments: Figure 4: regarding panels F and G, are there any significant differences among the different parameters evaluated? How many biological replicates were done? If so, please include the respective significances and statistical analysis in the methods section and discuss all these relevant aspects in the manuscript.
Author response: We have added all new results into the Figures 4 (comparing 3D vs. 2D) and Figure 5 (effects of TNF-α). These data represent results pooled from 3 biological replicates. As noted in the previous response, similar trends in results were found in the data across 3 independent device trials, indicating that the method is reproducible. When presenting JAnaP results in all of our publications, we choose to display all cells pooled to highlight the heterogeneity in junction phenotypes across cells; this decision was previously made in consultation with a statistician, Dr. Yuji Zhang, our collaborator at the University of Maryland Baltimore. Hence, we did the same here.
Reviewer comments: The junctional analysis evaluation by the new analytical method developed clearly presents advantages in providing a semi-automated quantitative evaluation of cell junction morphology, which reflects the barrier health/integrity/tightness among possible different treatments. In this regard, did the authors test compounds known to induce BBB leakage and/or to increase BBB tightness to support the usefulness of the model implemented? If so, it should be included; if not, it should at least be discussed the anticipated results in the main text. Moreover, could this method of analysis be applied/transposed to other BBB models besides the one described in this manuscript? And to other TJs and/or AJs proteins? Please discuss.
Author response: We thank the reviewer for this excellent suggestion. In new experiments, we added TNF-α into the 3D vessel and confirmed that this model can be used for evaluating changes in the barrier integrity under inflammatory conditions (Section 3.5 and Figure 5). Also, in the revised manuscript, we have added results for Occludin and Claudin-5. These new results are included in Sections 3.4 and 3.5 in the manuscript. Claudin-5 disappeared from iBMEC-like cells in the 3D vessels, while Occludin was present (Figure 5). Finally, we have added the results of the comparation between the 2D and 3D cells in Section 3.4 and Figure 4. Interestingly, we found significant differences between the ZO-1 junction phenotypes in 2D vs. 3D, with less continuous and more discontinuous junctions in the 3D vessels.
Generl aspects/typos:
Reviewer comments: Authors’ email – different fonts, different from the journal formatting. Please correct.
Author response: We have corrected the Author’s email to the journal formatting.
Reviewer comments: Main text – different formatting than the one required by the journal, with extra spacing. The same is true for figure captions. Please correct.
Author response: We have updated the formatting and the figure captions.
Reviewer comments: Grammar – consider revising English language before publication
Author response: We have revised the English language.
Reviewer comments: Line 86 – missing ref
Author response: This reference has been added.
Reviewer comments: Material and methods – commercial brands of some used reagents are missing (e.g. PDMS, UCT, etc). Please include.
Author response: The commercial brands of reagents used have been added.
Reviewer comments: Line 138 – RA states for ascorbic acid? Or retinoic acid? Please revise.
Author response: Thank you for pointing this out. We corrected the RA to retinoic acid.
Reviewer comments: Figure 4 – Figure caption only goes until E while the figure panels go until letter I. Please complete figure legend.
Author response: We have added new results to Figure 4 and updated the figure legends.
Round 2
Reviewer 1 Report
The new data comparing cylindrical versus flat cell culture is interesting and helps improve the quality of the MS.
Line 29: How do they assess barrier integrity?
There is still a problem with lines 728-732, which now read “Traditionally, trans-endothelial electrical resistance (TEER) is utilized for in vitro barrier function evaluation[48], but this technique does not easily translate to 3D microvessels, since the measurement of TEER from the ends of a long channel is impossible without access to the abluminal space[49].” There are two issues: 1) It is necessary to access to the abluminal space no matter how you measure TEER, ends or sides. 2) Assuming you have access to the abluminal space, measurement of voltage differences from the ends of a long tube are affected by the electrical-cable-like properties of the tube, as discussed in Odijk, which is now cited. The measurement of TEER of a tube requires a long wire electrode on the inside and a similarly sized one on the outside of the tube. It would suffice to say that “long electrodes are required in both the luminal and abluminal spaces.”
With regard to my previous comment “The absence of a perfused abluminal space can also lead to physiologically unrealistic gene expression, as shown by Shin et al., iScience 15, 391 – 406, May 31, 2019 ª 2019 https://doi.org/10.1016/, j.isci.2019.04.037,” the response misses the point. Yes, shear stress in the luminal space is important, but Shin showed that interstitial fluid flow may affect gene expression. Hence the solid wall formed by PDMS may produce an altered phenotype because abluminal factor may not be removed or metabolized. The authors need to discuss this problem. Read Shin carefully!
The response to my question about static culture, the authors deleted not one but two pieces of information: “It is not recommended to maintain a static culture of iBMEC-like cells more than Day 10, since part of the microvessels may be overgrown and detached from the microchannels for prolong culture in this model. Hence, this BBB chip model would be most useful for short timecourse experiments.” How are the cells kept alive in the tube without perfusion? I’m surprised that they lived for 10 days without a single media change, or was there slow flow between the reservoirs with different heights? The fact that the cells were not observed in shear makes the results even less exciting and less relevant.
Lines 196-201: What container was used for the differentiation? Was the “surface” a glass slide, a cover slip, a Petri dish, a well plate, or a flask?
Line 220 “Dow” not “DOW” It’s a company name, not an acronym.
Lines 239-240: I do not understand what exactly was done: “To secure the integrity of the two microchannel ends, an additional baking step employing PDMS was performed for sealing purposes. “ Were the open ends plugged with uncured PDMS and then baked? Figures 1 and 2 need to show this step.
Line 277 “coated” not “cocated”
Line 443: What does it mean to say “the PDMS was … polymerized again”? It can only be polymerized once.
Lines 677-681. Taking an injection-molded Nortis chip out of the box, adding the collagen, letting it set up, and pulling the fiber to create a channel could not be simpler and require less skill. It is a lot easier than making one’s own chip…
Lines 704-726. How do the effects of TNF-a compare to those from more conventional studies?
Author Response
Reviewer comment: The new data comparing cylindrical versus flat cell culture is interesting and helps improve the quality of the MS.
Reviewer comment: Line 29: How do they assess barrier integrity?
Author responses: In this study, we assessed the barrier integrity by quantitatively analyzing changes of junction phenotypes and coverages. We decided to change “barrier integrity” to “cell-cell junction integrity” to reflect the fact that we did not measure permeability, but rather junction phenotypes.
Reviewer comment: There is still a problem with lines 728-732, which now read “Traditionally, trans-endothelial electrical resistance (TEER) is utilized for in vitro barrier function evaluation[48], but this technique does not easily translate to 3D microvessels, since the measurement of TEER from the ends of a long channel is impossible without access to the abluminal space[49].” There are two issues: 1) It is necessary to access to the abluminal space no matter how you measure TEER, ends or sides. 2) Assuming you have access to the abluminal space, measurement of voltage differences from the ends of a long tube are affected by the electrical-cable-like properties of the tube, as discussed in Odijk, which is now cited. The measurement of TEER of a tube requires a long wire electrode on the inside and a similarly sized one on the outside of the tube. It would suffice to say that “long electrodes are required in both the luminal and abluminal spaces.”
Author responses: We thank the reviewer for this comment. We have rephrased the sentences in lines 532-533: “Traditionally, trans-endothelial electrical resistance (TEER) is utilized for in vitro barrier function evaluation[49], but this technique does not easily translate to 3D microvessels, since the measurement of TEER from the ends of a long channel is challenging and the long electrodes are required in both luminal and abluminal spaces[49].”
Reviewer comment: With regard to my previous comment “The absence of a perfused abluminal space can also lead to physiologically unrealistic gene expression, as shown by Shin et al., iScience 15, 391 – 406, May 31, 2019 ª 2019 https://doi.org/10.1016/, j.isci.2019.04.037,” the response misses the point. Yes, shear stress in the luminal space is important, but Shin showed that interstitial fluid flow may affect gene expression. Hence the solid wall formed by PDMS may produce an altered phenotype because abluminal factor may not be removed or metabolized. The authors need to discuss this problem. Read Shin carefully!
Author responses: We thank the reviewer for their comments. The limitation concerning the flow has been mentioned in the Discussion. In addition, we add a brief discussion about altered gene expression in response to fluid flow. Lines 568-571: “In our study, the solid wall formed by PDMS may produce an altered phenotype because abluminal factors may not be removed or metabolized in our system. Additionally, the flow rates have been shown to affect the gene expression[57]. The sustained cultivation of the model over an extended period remains arduous, particularly in the presence of flow, necessitating ongoing optimization of conditions.” Later in that same paragraph, we also added the following note: “However, it is still worthwhile to evaluate simplified BBB models to determine the minimum factors necessary to include in order to achieve behaviors that are predictive of in vivo outcomes.”
Reviewer comment: The response to my question about static culture, the authors deleted not one but two pieces of information: “It is not recommended to maintain a static culture of iBMEC-like cells more than Day 10, since part of the microvessels may be overgrown and detached from the microchannels for prolong culture in this model. Hence, this BBB chip model would be most useful for short timecourse experiments.” How are the cells kept alive in the tube without perfusion? I’m surprised that they lived for 10 days without a single media change, or was there slow flow between the reservoirs with different heights? The fact that the cells were not observed in shear makes the results even less exciting and less relevant.
Author responses: The cells were seeded into the microchannel on Day 6 of iBMEC differentiation. The immunostaining and the junction analysis were performed on Day 10. We cultured the cells in the microchannel for 4 days, during which the medium changes and the treatments were performed. The details have been outlined in the methods section. The timeline of the cell differentiation and vessel formation has been added to the schematic figure 1k.
Reviewer comment: Lines 196-201: What container was used for the differentiation? Was the “surface” a glass slide, a cover slip, a Petri dish, a well plate, or a flask?
Author responses: We used 6-well plates. We specified this information in Methods.
Reviewer comment: Line 220 “Dow” not “DOW” It’s a company name, not an acronym.
Author responses: We thank the reviewer pointing this out. We have corrected it.
Reviewer comment: Lines 239-240: I do not understand what exactly was done: “To secure the integrity of the two microchannel ends, an additional baking step employing PDMS was performed for sealing purposes. “ Were the open ends plugged with uncured PDMS and then baked? Figures 1 and 2 need to show this step.
Author responses: We thank the reviewer for this comment. The open ends were added with PDMS mix and then baked for sealing. Since we used a tiny amount of the PDMS to seal the end, it would not be possible to reveal this on the picture with appropriate scale. Instead, we described these steps in the methods section and clarified the text in the second revision.
Reviewer comment: Line 277 “coated” not “cocated”
Author responses: We have corrected this typo.
Reviewer comment: Line 443: What does it mean to say “the PDMS was … polymerized again”? It can only be polymerized once.
Author responses: We thank the reviewer for pointing this out. We have corrected it to “polymerized” and removed “again.”
Reviewer comment: Lines 677-681. Taking an injection-molded Nortis chip out of the box, adding the collagen, letting it set up, and pulling the fiber to create a channel could not be simpler and require less skill. It is a lot easier than making one’s own chip…
Author responses: We appreciate this suggestion from the reviewer. We may try this method later as we incorporate other factors, such as pericytes, astrocytes, other brain cells, ECM, and flow dynamics.
Reviewer comment: Lines 704-726. How do the effects of TNF-a compare to those from more conventional studies?
Author responses: We thank the reviewer for their comments. We have added the discussion in lines 521-525: “Conventionally, TNF-α induces loss in barrier properties by decreasing the expression of junction proteins, elevating permeability, and reducing the TEER in BMECs[47]. In agreement with these findings, our system exhibited sensitivity to TNF-α treatment, as we observed distinct alterations in ZO-1 and Occludin in iBMEC-like cells in the 3D vessels following exposure to TNF-α.”
Reviewer 2 Report
I would like to thank the authors for the excellent effort they have made to address the issues that I raised with their paper. With these additional experiments the paper now provides a more robust assessment of their method, along with some interesting insights into the difference between 2D and 3D cell culture cell-cell junction phenotypes. I feel that this work will be of interest to many researchers investigating endothelial barrier integrity, relevant to a number of diseases, and will make a valuable contribution to the field. I therefore recommend that the paper should be published in Bioengineering.
Author Response
We sincerely thank Reviewer 2 for taking the time to review our manuscript!